Rapid parameter estimation for selective inversion recovery myelin imaging using an open-source Julia toolkit

http://orcid.org/0000-0003-2091-868X Sisco Nicholas J. 1 2 3
http://orcid.org/0000-0003-0490-005X Wang Ping 1 2 3
http://orcid.org/0000-0003-1137-5003 Stokes Ashley M. 1 2 3
Dortch Richard D. 1 2 3 richard.dortch@barrowneuro.org
1 Department of Translational Neuroscience, Barrow Neurological Institute , Phoenix, AZ , United States
2 Barrow Neuroimaging Innovation Center, Barrow Neurological Institute , Phoenix, AZ , United States
3 St. Joseph’s Hospital and Medical Center , Phoenix, Arizona , United States of America
Gollo Leonardo
Electronic publication date: 2022 Mar 29
Publication date: 2022
Volume: 10
Electronic Location ID: e13043
Received 2021 Oct 1; Accepted 2022 Feb 10
Copyright: © 2022 Sisco et al.
Copyright year: 2022
Copyright holder: Sisco et al.
License: This is an open access article distributed under the terms of the Creative Commons Attribution License, which permits unrestricted use, distribution, reproduction and adaptation in any medium and for any purpose provided that it is properly attributed. For attribution, the original author(s), title, publication source (PeerJ) and either DOI or URL of the article must be cited.
License URL: https://creativecommons.org/licenses/by/4.0/

Keywords: MRI, Multiple Sclerosis, Julia Language, Quantitative MRI

Funding: National Institute of Neurological Disorders and Stroke R01: NS097821 National Center for Advancing Translational Sciences R21: TR003312 This work was supported by the National Institute of Neurological Disorders and Stroke (R01: NS097821) and National Center for Advancing Translational Sciences (R21: TR003312) at the National Institute of Health. The funders had no role in study design, data collection and analysis, decision to publish, or preparation of the manuscript.

==============================
Background

Magnetic resonance imaging (MRI) is used extensively to quantify myelin content, however computational bottlenecks remain challenging for advanced imaging techniques in clinical settings. We present a fast, open-source toolkit for processing quantitative magnetization transfer derived from selective inversion recovery (SIR) acquisitions that allows parameter map estimation, including the myelin-sensitive macromolecular pool size ratio (PSR). Significant progress has been made in reducing SIR acquisition times to improve clinically feasibility. However, parameter map estimation from the resulting data remains computationally expensive. To overcome this computational limitation, we developed a computationally efficient, open-source toolkit implemented in the Julia language.

Methods

To test the accuracy of this toolkit, we simulated SIR images with varying PSR and spin-lattice relaxation time of the free water pool (R1f) over a physiologically meaningful scale from 5% to 20% and 0.5 to 1.5 s−1, respectively. Rician noise was then added, and the parameter maps were estimated using our Julia toolkit. Probability density histogram plots and Lin’s concordance correlation coefficients (LCCC) were used to assess accuracy and precision of the fits to our known simulation data. To further mimic biological tissue, we generated five cross-linked bovine serum albumin (BSA) phantoms with concentrations that ranged from 1.25% to 20%. The phantoms were imaged at 3T using SIR, and data were fit to estimate PSR and R1f. Similarly, a healthy volunteer was imaged at 3T, and SIR parameter maps were estimated to demonstrate the reduced computational time for a real-world clinical example.

Results

Estimated SIR parameter maps from our Julia toolkit agreed with simulated values (LCCC > 0.98). This toolkit was further validated using BSA phantoms and a whole brain scan at 3T. In both cases, SIR parameter estimates were consistent with published values using MATLAB. However, compared to earlier work using MATLAB, our Julia toolkit provided an approximate 20-fold reduction in computational time.

Conclusions

Presented here, we developed a fast, open-source, toolkit for rapid and accurate SIR MRI using Julia. The reduction in computational cost should allow SIR parameters to be accessible in clinical settings.

Introduction

Conventional magnetic resonance imaging (MRI) techniques are exquisitely sensitive to pathology such as demyelination, edema, and axonal loss; however, they generally lack pathological specificity and are dependent on numerous acquisition parameters. As a result, there has been increased interest in quantitative MRI methods (Tabelow et al., 2019; Mancini et al., 2020) to derive indices with improved pathological specificity and reduced sensitivity to experimental parameters. In general, this requires the acquisition of multiple images with different experimental parameters. The signal in each voxel of the image series is then fit with the appropriate model—often via nonlinear least-squares methods—to estimate quantitative MRI parameters. Unfortunately, this process can be computationally expensive for high-resolution or large field-of-view applications such as whole-brain scanning.

One such MRI method is quantitative magnetization transfer (qMT) imaging, which provides indices (macromolecular pool size ratio or PSR) related to total myelin content in white matter (Mancini et al., 2020; van der Weijden et al., 2021). Despite the promise of quantitative myelin measurements, conventional qMT methods require specialized sequences and complicated analyses that are unavailable at most sites, limiting widespread adoption. We recently overcame the first of these limitations by developing a novel qMT method called selective inversion recovery (SIR), which uses inversion recovery sequences that are available on most clinical MRI scanners. We demonstrated that the resulting PSR values are repeatable across scans and relate to myelin content, as well as disease duration and disability in multiple sclerosis (MS) (Dortch et al., 2011, 2013; Bagnato et al., 2020). We later optimized SIR sampling schemes and acquisition readouts to ensure clinical applicability (Dortch et al., 2018; Cronin et al., 2020). Together, these studies demonstrated that whole-brain SIR data could be acquired in under 10 min.

Despite these methodological improvements in acquisition, widespread SIR adoption is currently hindered by long computation times required to estimate model parameters, which can take on the order of tens of minutes (depending on the specifics of the hardware) for whole-brain acquisitions using our current MATLAB implementation. These long computation times stem from the requirement to fit each voxel to the biexponential SIR signal model using nonlinear regression methods, which can be computationally expensive. This is exacerbated in whole-brain scans, where the fit is performed for each voxel independently, resulting in >100,000 total regressions to estimate whole-brain parametric maps. As a result, faster computational techniques are needed to foster widespread clinical adoption of SIR. In addition, techniques that are composable, dynamic, general-purpose, reproducible, and open-sourced would further minimize barriers related to code sharing and adoption.

A relatively new language named Julia fits all these requirements. Julia works on all major operating systems—Windows, MacOS, and Linux—and has quickly situated itself as a computational tool capable of reaching petaFLOPS performance (Claster, 2017). As such, it has been used in diverse computationally intensive fields ranging from earth astronomical cataloging (Regier et al., 2018) to quantitative MRI (Smith et al., 2015; Doucette, Kames & Rauscher, 2020). Currently, many MRI processing tools are developed using MATLAB (Ashburner et al., 2013) and Python (Smith et al., 2004; Gorgolewski et al., 2011), which have well-known limitations shared by other interpreted languages, most notably longer execution times. Julia has an intuitive user interface, is similarly portable and readable to MATLAB and Python, and retains most of the functionalities and syntax their users recognize (Perkel, 2019). However, since Julia is compiled at run time, it has inherent qualities that make it more computationally efficient, thus allowing it to approach C/C++-like speeds (Bezanson et al., 2017, 2018). In other words, Julia strikes a balance between syntax that looks like an interpreted language, e.g., Python, R, MATLAB, etc., but runs with computational efficiency like a compiled language.

The goal of this work was to develop and validate an open source, free, fast, flexible, and simple Julia toolkit for estimating SIR parameter maps. More specifically, we developed a Julia-based toolkit for rapid SIR parameter estimation that resulted in a 20-fold reduction in computational time over our previous MATLAB implementation. We evaluated this toolkit on simulated SIR images and high-resolution images collected from tissue-model phantoms and a healthy volunteer. Since our code is freely available and easily portable, we anticipate this toolbox will be widely accessible to researchers and clinicians to efficiently and accurately obtain SIR parameters. In addition, the toolkit is developed in a modular nature, allowing it to be easily extended to other nonlinear regression problems common in quantitative MRI applications.

Methods

Theory

Selective inversion recovery (SIR) imaging (Edzes & Samulski, 1977; Gochberg & Gore, 2003, 2007) is based on a low-power, on-resonance inversion pulse that inverts the longitudinal magnetization (Mzf) of free water protons with minimal perturbation of magnetization (Mzm) for protons in the macromolecular pool. Whereas traditional inversion recovery sequences use a pre-delay time tD = 5 × T1 (defined as the time from the center of the last spin echo in the readout until the next inversion pulse) to ensure full recovery before each inversion, SIR methods often use reduced tD to yield gains in efficiency, based on the assumption that both pools are saturated at tD = 0 (Gochberg & Gore, 2007; Cronin et al., 2020). Mathematically, we can describe the resulting time evolution of the longitudinal magnetization vector Mz=[MzfMzm]T(Dortch et al., 2013, 2018) as

(1) Mz(tI,tD)=[eAtIS(I−eAtD)+(I−eAtI)]M0

where tI is the inversion time, S=diag(Sf,Sm) accounts for the inversion pulse effect on each pool (Sf = −1 and Sm = 1 indicate complete Mzf inversion and no Mzm saturation, respectively), I is the identity matrix, M0=[M0f M0m]T is the equilibrium magnetization vector, and A is a matrix with components

(2) A=[−(R1f+kfm)kmfkfm−(R1m+kmf)]

Here R1f,m are the spin-lattice relaxation times of each pool and kmf is the exchange rate from the macromolecular to free pool. Given dynamic equilibrium and static compartment sizes, the exchange rate in reverse direction can be stated as kfm=PSR×kmf. For free water protons, the observed SIR signal is directly proportional to the Mzf component in Eq. (1), which can written algebraically as a biexponential function with respect to tI (Dortch et al., 2011).

This results in a model with seven independent parameters: PSR, R1f, R1m, Sf, Sm, M0f, and kmf. Several assumptions can be made to reduce model parameters during fitting. Sm can be numerically estimated as Sm = 0.83 ± 0.07, assuming a 1-ms hard inversion pulse, Gaussian lineshape, and T2m = 10–20 µs (Dortch et al., 2011). In addition, the SIR signal is relatively insensitive to R1m; therefore, it is often assumed that R1m = R1f (Li et al., 2010). Furthermore, kmf was shown to be relatively consistent within normal (kmf = 12.5 s−1 for human brain) and diseased neural tissue, and optimized SIR acquisitions have been developed to minimize bias in other parameters estimates (e.g., PSR, R1f) when an assumed kmf values is used (Dortch et al., 2011, 2018). This results in a model with four independent parameters (PSR, R1f, Sf, and M0f), which can be estimated via nonlinear regression of SIR data acquired at four (or more) different tI and/or tD values with the biexponential function shown in Eq. (1).

Julia implementation

For our Julia implementation, nonlinear regression was performed using curve_fit from the LsqFit.jl package, which is an implementation of the efficient Levenberg–Marquardt algorithm. The only non-default parameter for our fitting routine was the use of automatic forward differentiation rather than the default central differencing, which has been shown to speed up Jacobian estimation at little cost to parameter estimation accuracy (Revels, Lubin & Papamarkou, 2016).

Julia has several unique features that were exploited to maximize both the efficiency and usability of our toolkit. First, multithreading is supported by Julia and is easily implemented by appending the @threads macro to any for-loop call. In our toolkit, this was appended to the for-loop used to loop over regressions for each voxel. In contrast to MATLAB, for-loops are generally encouraged in Julia rather than using vectorized code, as the former often yields highly efficient machine code. In the present implementation, we provided the option to either define certain parameters (e.g., Sm and R1m) or use a default value if no argument is passed. Finally, the dispatch of methods in Julia can be associated with multiple input variable types, which yields code that is simultaneously flexible and efficient. In our case, this allowed for the dispatch of different SIR fitting models simply based on whether kmf was provided as an input (assumed kmf) or not (estimated kmf).

Simulation studies

To evaluate the SIR Julia toolkit, SIR data were simulated using pulse sequence parameters (tI: 15, 15, 278, and 1,007 ms and tD: 648, 4,171, 2,730, and 10 ms) that correspond to the optimized experimental parameters (Dortch et al., 2018) used in our phantom and whole-brain scans. Simulated PSR and R1f values were linearly varied from 5–25% and 0.5–1.5 s−1, respectively, over a 128 × 128 grid to cover the full range of values observed in human white matter at 3.0 T. Sf and M0f were held constant at −1 and 1, respectively, since these parameters are not biologically relevant. Rician noise was added to the image at each tI and tD to generate noisy data with a signal-to-noise ratio (SNR) of 250 relative to M0f. This produced a final simulated dataset with 128 × 128 × 4 matrix dimensions, where the final dimension represents the data acquired at each combination of tI and tD.

Fits for each simulated voxel were then performed using our Julia toolkit on a Dell Precision® Mobile Workstation 7750 with Intel® Comet Lake Core™ i9-10885H vPRO™ @ 2.4 GHz CPU with Hyper-threading® enabled (eight physical cores, 16 logical cores), and 16 GB non-ECC DDR4 RAM at 2,933 MHz using Ubuntu 20.04.2 LTS through Windows Subsystem Linux. The code generated here was additionally evaluated on Windows 10 (Dell Precision detailed above) and an iMac (Intel® Kaby Lake™ i7-7700K @ 4.2 GHz CPU with Hyper-threading® enabled (four physical cores, eight logical cores), 32 GB non-ECC DDR4 RAM at 2,400 MHz, running MacOS Catalina 10.15.7). The code was tested on Julia 1.5.2 and Julia 1.6.2, and both versions completed without error.

Phantom studies

Bovine serum albumin (BSA, Sigma-Aldrich, St. Louis, MO, USA) phantoms were created in 50-mL conical vials by first solubilizing BSA in 15 mL of ddH2O (18.2 MΩ·cm at 25 °C, double-distilled water) until fully dissolved, followed by adding ddH2O up to a final volume of 30 mL after accounting for glutaraldehyde (Electron Microscopy Science) volume for final BSA concentrations equal to 20%, 10%, 5%, 2.5%, and 1.25% (w/v). The vials were centrifuged at 3,500×g for 10 min to reduce bubbles before the crosslinking reaction. Glutaraldehyde was added to a concentration of 12% from a 50% glutaraldehyde stock in ddH2O. Once the glutaraldehyde was added, the mixture was gently mixed to avoid bubble formation, centrifuged again with the same settings as above, and allowed to react overnight at 4 °C. To more directly investigate the relationship between BSA concentration and our SIR measures, we converted PSR to reflect the fraction of macromolecular to free water magnetization using the following expression: f=M0mM0m+M0f=PSR1+PSR.

MRI was performed using a 3.0T Ingenia™ (Philips®, Amsterdam, Netherlands) scanner with a dedicated 32-channel head coil. The phantoms were placed in a plastic 50-mL conical tube holder and positioned in the center of the RF coil. The same tI and tD used for simulations were used for phantom scanning. SIR data were collected at bore temperature with an inversion recovery prepared 3D turbo spin-echo (TSE) sequence. The field of view (FOV) was set to 120 × 120 × 30 mm3, with 0.5 × 0.5 × 3.0 mm3 resolution, matrix size of 240 × 240 × 10, echo time (TE): 96 ms, TSE factor of 22, and compressed sensing acceleration factor of 8 (Wang, Sisco & Dortch, 2021). The resulting data were fit using our Julia toolkit as described above for the simulated data using a fixed kmf = 35.0 s−1 based on previous SIR experiments in BSA phantoms (Dortch et al., 2018).

Whole brain human studies

To test the clinical applicability of our code, analogous SIR data were collected, and parameter maps estimated performed in a healthy volunteer (36-year-old, male). All scanning parameters were identical to the phantom scans except: FOV of 210 × 210 × 90 mm3, acquired isotropic resolution of 2.25 mm3, with reconstructed matrix size of 224 × 224 × 40 and reconstructed resolution of 0.94 × 0.94 × 2.25 mm3, and TE: 65 ms. Preprocessing of the human SIR data was performed with FSL (https://fsl.fmrib.ox.ac.uk/fsl/) (Smith, 2002) and included rigid registration using FLIRT to correct for motion and brain extraction using BET. During fitting, kmf was fixed to the mean value reported in healthy human brain at 3.0 T (kmf = 12.5 s−1). This study was performed, including written consent, per the St. Joseph’s Hospital and Medical Center Institutional Review Board (IRB number PHX-22-500-006-30-08).

Statistical analysis

We evaluated the accuracy and precision of the parameter estimates (PSR and R1f) generated by our Julia toolkit relative to the simulated values via histogram analyses and Lin’s concordance correlation coefficients (LCCC). All statistical analyses were performed using R, and the package epiR (Stevenson & Sergeant, 2021) was used for LCCC estimation.

Code usage examples

To encourage the use of the Julia toolkit, we provide easy-to-use bash-shell code that can be copied line by line and used within a Linux-like command line or saved as a script for execution in our GitHub repository (https://github.com/nicksisco1932/The_SIR-qMT_toolbox). Additional documentation and source code can is also provided in this repository. Required input parameters include the SIR images in either NIfTI or MATLAB’s MAT format along with arrays for inversion and predelay times. Optional parameters can also be defined for kmf, Sm, and R1m, depending on the application; otherwise, the default values listed above are used. Alternatively, the toolkit can be implemented as a shell script in bash or can be incorporated into Python and MATLAB pipelines. Finally, we supply a Jupyter notebook tutorial written for Julia to create and evaluate the simulation data shown in this manuscript. This notebook, along with code snippets needed to run our Julia toolkit via Python, MATLAB, bash scripts, or the command line, can all be found at our code repository.

A separate challenge that is common in quantitagive neuroimaging analysis relates to image format and data loading. To provide flexibility for other imaging formats (aside from NIfTI and MAT files), a code branch called using_pycall_import was developed to enable the usage of the Python package nibabel, which imports nibabel software (Brett et al., 2020) to read in various types of medical images, such as DICOM and PARREC (Philips format), as well as NifTI. However, as this branch implementation requires a Python environment with nibabel installed, it was implemented separately to simplify usage for end users.

Results

In Fig. 1, we show the simulated and fit R1f (Fig. 1A) and PSR (Fig. 1B) values for each pixel, along with the residuals from the simulated and fit data for R1f (Fig. 1C) and PSR (Fig. 1D). The difference between simulated and estimated R1f (Fig. 1C) and PSR (Fig. 1D) showed no systematic differences. Quantitatively, these data are nearly identical to the known values (Figs. 2A, 2B) with LCCC = 0.99/0.99 and RMSE = 2.2%/9.2% for R1f/PSR shown in Figs. 2C and 2D. Figures 1 and 2 support the accuracy of the Julia toolkit over a range of biologically realistic values in the presence of experimental noise.

Figure 1 Simulated and Fit SIR images.

The simulated images were generated with constant inversion times of 15, 15, 278, and 1,007 ms and delay times of 648, 4,171, 2,730, and 10 ms with PSR and R1f changing per pixel in a 128 × 128 matrix and Rician noise added, depicted in the A and B. We fit the simulated image to the SIR-qMT model, resulting in the central panel parameter map for A and B. The difference between the simulated image and the parameter map is depicted in C and D. Qualitatively, C and D show that R1f and PSR were estimated with high accuracy relative to the simulated values with LCCC = 0.99/0.99 and RMSE = 2.2%/9.2% for R1f/PSR (the distribution of the differences is assessed in Figs. 2A, 2B).

Figure 2 Probability density histograms and Lin’s Concordance Correlation Coefficient plots.

Simulated phantoms were fit with high agreement and precision. Percent differences between the fit and known data for R1f values (A) and PSR values (B) have small, differences which is explained by Rician noise as expected. In C and D, the PSR and R1f show high agreement between the fit and simulated values with LCCC = 0.99 and 0.98, respectively, while the solid line for unity and dotted correlation fit are nearly overlapped. These data give us confidence that our Julia code is fitting the data to the expected values.

Next, we performed real-world SIR experiments to test our Julia toolkit in samples with well-characterized PSR and R1f values using BSA phantoms. The values from the fit are displayed in Fig. 3 and correspond to within 10% margin of error of published values of similar phantoms (Dortch et al., 2018). Figures 3A and 3B show the PSR and R1f values, respectively. The arrangement and percentage labels of BSA are depicted in Fig. 3C. The linear relationship between the SIR-derived f values and BSA concentration is shown in Fig. 3D with an intercept close to zero (0.003) and slope of 0.64 (standard error 0.002 and 0.019, respectively).

Figure 3 Tissue model phantom images.

Five BSA phantoms were used to assess the Julia model fitting shown here. BSA concentrations ranged from 1.25% to 20% (w/v). In A, PSR values were 0.9 ± 0.7, 1.9 ± 1.5, 3.9 ± 1.1, 6.5 ± 1.1, and 13.2 ± 2.9 as a function of BSA concentration. R1f shown in B values were 0.41 ± 0.01, 0.46 ± 0.03, 0.51 ± 0.02, 0.63 ± 0.002, and 0.83 ± 0.05 per BSA concentration. The BSA concentrations are depicted in C showing the arrangement of the phantom tubes when in the scanner. A black box marks the slice location that was that can be seen visually after a rotation. The values fit in these phantoms are similar to those found in literature using SIR within a 10% margin of error. Additionally, when PSR is converted to a fraction of macromolecular pool to free water, see Methods, it correlates well with BSA concentration with a near 0 offset, as expected. Deviations are likely due to scanner differences and minor phantom preparation method differences. The macromolecular to free rate constant (kmf) was held constant at 35.0 s−1 for phantom fitting.

Lastly, we tested our Julia code using whole-brain data from a healthy volunteer, as shown in Fig. 4. Figure 4A shows the raw image at tI,tD = 278, 2,730 ms; Fig. 4B shows the expected contrast from PSR maps with higher values in white matter; Fig. 4C is the R1f map with higher values in the white matter; and Fig. 4D reflects inversion efficiency, which is characteristically flat (average Sf = −0.86 ± 0.14 for whole brain) and accounts for nonideal inversions of the water signal.

Figure 4 Representative SIR on a healthy volunteer.

Panel A represents the first data point corresponding to tI, tD = 278, 2,730 ms. B, C, and D are maps from the fit parameters pool size ratio, R1f, and Sf (B1 inhomogeneity), respectively. These images are consistent with published parameters, white matter have the highest relative PSR and R1f, while Sf remains relatively flat at 3T with slight increases near the posterior of this map.

For comparison, we evaluated the same whole brain with our original code written in MATLAB and generated identical maps, and Julia exhibited a significant reduction in computation time. More specifically, using the Core™ i9 laptop listed the Methods section and single threaded operations, MATLAB and Julia fit the entire brain (596,389 voxels within the brain mask) in 1,254 s (MATLAB) and 14 s (Julia); corresponding to an ~90× reduction in computation time for Julia. Using MATLAB parallel processing (parfor) improved performance for MATLAB to 224 s, but this was still approximately 16× slower than Julia single threaded operations and requires significant overhead related to initiating separate MATLAB processes. The Julia multi-threading macro requires substantially less overhead than MATLAB; however, it only marginally reduced computation times over Julia single-threaded operations in our current implementation, suggesting that memory allocation may be the rate limiting factor in our Julia code. For our software implementation and hardware (described above), dual threading yielded the largest reduction in computation time relative to single threading (25–30% reduction); however, this is likely hardware dependent and single threading via Julia was found to be exceedingly fast. A more intuitive way to compare computational times is to measure how many voxels were fit per second, which was 42,714 and 2,662 voxels per second for Julia’s and MATLAB’s fastest times, respectively.

Discussion

We present an efficient implementation of SIR parameter estimation using the Levenberg–Marquardt algorithm via the Julia language. We used this toolkit to estimate SIR parameters on simulated data, BSA phantoms, and whole-brain human data. We then tested the run time of our toolkit to fit whole-brain SIR images resulting in PSR maps fit in 14 s for using Julia, which took MATLAB 224 s (using parallel processing), amounting to a nearly 16-fold decrease in computational time. Additionally, we note that the entire script, including reading and writing files, as well as fitting, takes only 29 s to complete. The robustness of the fit was evaluated using the simulated data with Rician noise added (Figs. 1A and 1B). After fitting, the residuals from the known data were characteristic of the noise encoded in the simulated image (Figs. 1C and 1D) with very high correlation according to LCCC, i.e., the fit recovered the data with high precision and accuracy (Fig. 2). Next, we acquired SIR data on a 3T scanner using phantoms made with BSA, and the estimated PSR and R1f parameters agreed with previously published data (Dortch et al., 2018) (Fig. 3). The linear relationship between f and BSA, shown in Fig. 3D, along with the near zero offset provides good evidence that the phantoms were consistent, and that the fitting code performs well with real-world data. Finally, we acquired whole-brain data on a healthy volunteer at 3T, which showed that SIR parameters were consistent with expectations. More specifically, the PSR values (Fig. 4B) and R1f (Fig. 4C) were higher for white matter and consistent with published values (Dortch et al., 2018) that used our previous MATLAB implementation, while Sf was relatively flat (Fig. 4D).

Due to its computational efficiency, Julia has become an increasingly popular tool for use in MRI data analysis. For example, it has been used for fitting dynamic contrast-enhanced MRI (DCEMRI.jl) data in less than a second (Smith et al., 2015) and myelin water imaging (MWI) with Decomposition and Component Analysis of Exponential Signals (DECAES.jl) that showed 50-fold improvement in computational time (Doucette, Kames & Rauscher, 2020). We should note that numerous other MRI computational packages exist for quantitative analysis, including QUIT (QUantitative Imaging Tools, Wood, 2018) and qMRLab (Quantitative MRI analysis, under one umbrella, Karakuzu et al., 2020), with the former written predominately in C++ and Python and the latter written in MATLAB. In particular, qMRLab has a large array of tools for processing MT data, including a SIR-FSE fitting routine that is similar to our previous MATLAB implementation.

In the present study, our toolkit showed whole-brain SIR data can be fit with a biexponential model in clinically feasible time of less than half a minute using a high-end laptop with a virtualized Linux operating system within the Windows 10 system and on the same laptop using Windows 10 version of Julia. Additionally, the toolkit was equally efficient on a standard desktop computer running MacOS. Given that our toolkit is highly efficient on all operating systems, easy to use, lightweight, and open source, we believe this opens the possibility of incorporating this toolset on any scanner operating system to significantly expand the clinical use for SIR. As the computational steps represents a barrier to the clinical implementation, we anticipate that the Julia-based implementation of SIR fitting is a critical step toward broader clinical use.

The implementation of Julia shown here is also a steppingstone for more comprehensive Julia computational implementation within the magnetic resonance research community. The fast and composable nature of our Julia toolkit allows additional model functions to be added with little effort. For example, we anticipate using our basic code design in other non-linear fitting models, such as for rapidly estimating T1, T2*, and T2 in other applications. Overall, a robust, easily adaptable, and fast computational tool would be a welcome addition to the field.

One limitation to the adoption of Julia stems from the fact that it is a relatively new language and is continuously being updated. This novelty can make the developed packages obsolete relatively quickly; however, the upside is that Julia versions greater than 1.0 are increasingly stable and are constantly improving with a dedicated and vibrant community of developers. For example, we chose to use established tools (FSL) for preprocessing steps (registration, segmentation) rather than develop them natively in Julia. We anticipate future work will focus on porting these tools into Julia, which would alleviate potential dependencies issues that can arise from using multiple software packages within a processing pipeline. Furthermore, GPU accelerated computing is continually expanding in Julia with the JuliaGPU.jl package (Besard et al., 2019), which does not require a specific brand of graphics card and could make GPU acceleration more accessible and fits even faster. We assessed the code presented here with two different versions of Julia and found no bugs or code failures in anticipation of this deprecation issue. Julia is highly flexible and can be easily adapted to suit the function of the user. Although we focused on standard model assumptions (fixed kmf, R1m = R1f), the flexibility of our Julia implementation allows one to alter these assumptions for each specific application. For example, kmf may be altered by inflammation (Harrison et al., 2015). Furthermore, increasing evidence suggests that R1m values are much slower than previously assumed, and these incorrect assumptions may bias R1f estimates (Wang et al., 2020). We believe that the combined flexibility and efficiency of our toolkit will allow investigators to systematically evaluate the impact of these model assumptions on estimated SIR parameters and, ultimately, deploy SIR as a clinical myelin biomarker.

Conclusions

We developed a fast, open-source toolkit for SIR MRI analysis using Julia. This toolkit was validated using simulations, phantoms, and healthy volunteer images. More specifically, myelin-related SIR parameters were estimated in simulated images with high accuracy and precision, agreeing with published values in tissue-mimicking phantoms. Whole-brain SIR myelin maps further demonstrated with a 20-fold reduction in computational time, providing evidence that this toolkit would be instrumental in a clinical setting.

We acknowledge Philips Healthcare and the Barrow Neurological Foundation.

Additional Information and Declarations

Competing Interests

Author Contributions

Human Ethics

Data Availability

The authors declare that they have no competing interests.

Nicholas J. Sisco conceived and designed the experiments, performed the experiments, analyzed the data, prepared figures and/or tables, authored or reviewed drafts of the paper, and approved the final draft.

Ping Wang performed the experiments, authored or reviewed drafts of the paper, and approved the final draft.

Ashley M. Stokes conceived and designed the experiments, analyzed the data, authored or reviewed drafts of the paper, and approved the final draft.

Richard D. Dortch conceived and designed the experiments, analyzed the data, authored or reviewed drafts of the paper, and approved the final draft.

The following information was supplied relating to ethical approvals (i.e., approving body and any reference numbers):

St. Joseph’s Hospital Institutional Review Board approved the study (IRB number PHX-22-500-006-30-08).

The following information was supplied regarding data availability:

The code is available at GitHub: https://github.com/nicksisco1932/The_SIR-qMT_toolbox.

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
