# Peer review of "Rapid parameter estimation for selective inversion recovery myelin imaging using an open-source Julia toolkit"

_PeerJ, doi:10.7717/peerj.13043_

## Round 0.1 · original submission · Major Revisions

Your manuscript has now been seen by two reviewers. You will see from their comments below that while they find your work of interest, some major points are raised. We are interested in the possibility of publishing your study, but would like to consider your response to these concerns in the form of a revised manuscript before we make a final decision on publication. We therefore invite you to revise and resubmit your manuscript, taking into account the points raised. Please highlight all changes in the manuscript text file.

·

Basic reporting

This paper is well written and clear.

1 - The paper does not set itself properly in the context of existing literature. Importantly, it does not cite the original qMRLab paper (Karakuzu, A., Boudreau, M., Duval, T., Boshkovski, T., Leppert, I. R., Cabana, J. F., … & Stikov, N. (2020). qMRLab: Quantitative MRI analysis, under one umbrella. Journal of Open Source Software, 5(53), 2343. doi: 10.21105/joss.02343). qMRLab provides an SIR FSE fitting routine and I think a comparison of the two would be interesting to readers (https://qmrlab.readthedocs.io/en/master/qmt_sirfse_batch.html).

1b - Other toolboxes which do not include SIR FSE, such as QUIT (Wood, (2018). QUIT: QUantitative Imaging Tools. Journal of Open Source Software, 3(26), 656, https://doi.org/10.21105/joss.00656), are of relevance to the discussion of toolboxes and languages.

2 - With regards to the SIR model itself, the recent work of van Gelderen (https://linkinghub.elsevier.com/retrieve/pii/S1053811920301877) is not discussed. van Gelderen calls into question the assumption that R1m=R1f, and instead indicates that R1m is approximately 4 per s at 3T. The authors should discuss how this contrasts with their fitting assumptions and would affect their results. Furthermore, papers such as https://linkinghub.elsevier.com/retrieve/pii/S0006322314007501 suggest that the exchange rate can change due to pathology (specifically inflammation), which suggests that the assumption of fixed kmf may not be safe.

3 - Much extraneous detail about the Julia language are included in the paper, which could be replaced by a reference or link to the Julia educational materials. For instance, I would remove the mention of multi-dispatch in the discussion, as I doubt any readers who have not previously used Julia will know what this is (and neither is this paper the correct place to explain the concept). It is unnecessary to give the specific names of code files or Julia functions. It is sufficient to only cite the papers that these are from. Similarly, the sentence on choice of editor should be removed. While I also use Visual Studio Code, it is of little relevance to the reader.

4 - Throughout the paper citations often appear after the full stop of the sentence they refer to, i.e. . (citation). This should be corrected.

Experimental design

The methods are described with sufficient detail and the analysis code is open source.

5 - The research question is not particularly well defined, and this is the central weakness of the paper as currently written. The core message of the paper is that the authors took an existing method and implemented the analysis in a new programming language which was faster. While the speedups are impressive, there is not significant new scientific content in the paper.

There are then two other stated goals - one is implementing the method directly on the scanner, and the other is the extension of their code into a toolbox for multiple techniques. However, these goals are merely aspirational and are not actually addressed in this paper. I think the passages about these goals should be removed or de-emphasised, and more emphasis given to why the work presented in the paper is interesting in its own right.

Validity of the findings

The main findings are valid.

6 - With reference to point 4, the authors do not discuss the numerous practical difficulties for implementation on the scanner such as motion correction and integration with the scanner image database. Motion correction in particular is not currently discussed in the paper at all. As in point 5, I suggest the passages about implementing the method on the scanner are removed or significantly rewritten as the authors to address such issues.

7 - It is not clear to me why the authors report the run time with differing numbers of Julia threads. It appears that performance decreases with an increasing number of threads, which is counter-intuitive. Furthermore it is not clear what is meant by "only improved fits by a small amount" on line 257. Both points should be clarified.

8 - The noise analysis in figures 1 & 2 is an interesting way of visualising the performance across a range of parameter values simultaneously. However, a diverging color map should be used for figure 1 C/D, and I suggest that the differences are scaled into fractions or percent, i.e. the presented variable should be (Fitted - Truth) / Truth. Currently due to the grayscale colorbar and use of absolute units, it is difficult to asses how the noise varies with parameter values.

9 - While the authors discuss the overhead of various aspects of Matlab, they do not discuss the overhead from Julia's Just-In-Time compilation. What was the start-up time of their code if running as a command-line script rather than in an interactive Julia session?

Additional comments

Overall this paper is well written, contains no substantive errors, and I think will be suitable for publication after addressing the points above. As currently written there is too much emphasis on the specifics of the Julia language itself, for which the interested reader can be directed to the educational materials, and also on potential future uses of the toolbox for which this paper does not actually present evidence of. The paper will be improved if more focus is given to the actual results and better comparisons are made to existing work.

·

Basic reporting

The authors reported an open-source and julia-based software package that can fit the biexponential MRI signals obtained from selective inversion recovery MRI scans, producing quantitative maps reflecting bound water fraction (with very highly significant implication to research and clinical studies of neuro-degenerative disorders). Credits should be given to authors for their great efforts of implementing Julia based MRI processing codes, building jupyter-notebooks (that greatly benefit students and trainees), and providing software interface for Matlab and Python users.

Experimental design

The study includes numerical simulation (including Rician noise that is appropriate for simulating MRI signals), phantom and human MRI. Overall it is well designed and rigorous.

Validity of the findings

The experimental results from simulation, phantom, and human studies demonstrate that the implemented Julia software package is valid, with a significantly reduced computation time as compared with their previous matlab implementation. Although human data were obtained from only 1 subject, this is sufficient for validating the implemented julia software.

Additional comments

As an MRI researcher with experience in julia programming, I am convinced that the reported study and the implemented open-source software will greatly benefit neuroimaging community. It would be great if the authors could further discuss the following in their manuscript.

First, when using the implemented software procedure in human studies, it is expected that the motion artifact should be addressed before fitting the biexponential signals. Could the authors discuss if the motion artifact should be removed with Julia (in future implementation) or other tools (e.g., FSL)?

Second, the readers of this paper might be interested to know if the program could be further accelerated with GPU computation. It is my understanding that GPU computation is also available in Julia, and I would be interested to know authors' opinion on this topic.

---

## Round 0.2 · Minor Revisions

Your manuscript has now been seen by the reviewers. You will see from the comments below that some constructive points must be considered. We therefore invite you to revise and resubmit your manuscript, taking into account these points. Please highlight all changes in the manuscript text file.

·

Basic reporting

The authors have addressed my concerns on reporting and I have no further comments.

Experimental design

The authors have addressed my concerns on design and I have no further comments.

Validity of the findings

I thank the authors for clarifying the results section of the paper in response to my previous comments. First, two minor points:

1 - The sentence at lines 250/251 of the paper, as written, suggests that the MATLAB code was faster when the opposite is true. I suggest this sentence is reworded.
2 - The authors do not appear to state the number of cores/threads available on the CPUs used for processing. I believe that 8 cores/16 hyper-threads are available for the serial numbers listed, which makes some sense given the ~6 times speed-up in MATLAB when enabling threading. These numbers would be helpful to the reader.

And now a major point. I cloned and ran the author's repository to examine the multi-threading issue earlier. I am not an expert in Julia, so I followed the instructions on multi-threading at this web-page: https://docs.julialang.org/en/v1/manual/multi-threading/

I noted that the author's main script, "run_me.sh", did not have the `--threads=auto` command-line argument in the Julia call.

I first ran the script as-is, and confirmed the authors observation that adding the @threads decorator did not change the run-time. I then added a print statement to output the number of available threads, which was 1. I then added the `--threads=auto` option and re-ran. The program then reported that 6 threads were available (on my 6 core machine), I confirmed that the program used all the cores by observing the process monitor, and this led to a 2x speed-up in program execution.

I have submitted the required changes as a pull-request on Github https://github.com/nicksisco1932/The_SIR-qMT_toolbox/pull/5

I ask the authors to confirm if they can replicate this behaviour on their machines. If they can, then the results section will require re-writing. If they cannot, I think they must provide a comprehensive explanation of the threading behaviour in Julia.

Additional comments

No further comments.

---

## Round 0.3 · accepted · Accept

Thank you for the detailed response letter. We are delighted to accept your manuscript for publication.

·

Basic reporting

The authors have addressed my concerns on reporting and I have no further comments.

Experimental design

The authors have addressed my concerns on reporting and I have no further comments.

Validity of the findings

I thank the authors for thoroughly responding to my request in the previous review. The behaviour of multi-threading in Julia is clearly more complex than is perhaps expected, but I agree with the authors that this is beyond the scope of the paper. I have no further issues with their work.

Additional comments

The authors have addressed my concerns on reporting and I have no further comments.